# Linear Opinion Pooling for Uncertainty Quantification on Graphs

**Clemens Damke**[1]                    **Eyke Hüllermeier**[1,2]

[1]Institute of Informatics, LMU Munich, Germany
[2]Munich Center for Machine Learning (MCML)

## Abstract

We address the problem of uncertainty quantification for graph-structured data, or, more specifically, the problem to quantify the predictive uncertainty in (semi-supervised) node classification. Key questions in this regard concern the distinction between two different types of uncertainty, aleatoric and epistemic, and how to support uncertainty quantification by leveraging the structural information provided by the graph topology. Challenging assumptions and postulates of state-of-the-art methods, we propose a novel approach that represents (epistemic) uncertainty in terms of mixtures of Dirichlet distributions and refers to the established principle of linear opinion pooling for propagating information between neighbored nodes in the graph. The effectiveness of this approach is demonstrated in a series of experiments on a variety of graph-structured datasets.

## 1 INTRODUCTION

Quantifying the uncertainty of predictions made by machine learning models is critical for applications where safety is important and mistakes can be costly. When assessing the uncertainty of a model's prediction, it is common and often useful to distinguish between two types of uncertainty: *Aleatoric uncertainty* (AU) arises from the stochasticity inherent to the data-generating process, and cannot be reduced by sampling additional data. For example, when tossing a fair coin, the outcome is uncertain, and this uncertainty is of purely aleatoric nature. *Epistemic uncertainty* (EU), on the other hand, is due to a lack of knowledge about the data-generating process; assuming that an appropriate model class is chosen, EU can be reduced by collecting more data and vanishes in the limit of infinite data [Hüllermeier and Waegeman, 2021]. For example, the lack of knowledge

about the bias of a coin is of epistemic nature, and it increases the (total) uncertainty about the outcome of a coin toss. This uncertainty, however, can be reduced by tossing the coin repeatedly and estimating the bias from the outcomes.

In the context of graph-structured data, *uncertainty quantification* (UQ) is particularly challenging due to the structural information as an additional contributing factor to the uncertainty. In this paper, we will focus specifically on the problem of UQ for (semi-supervised) node classification. Applications of this problem include, for example, the classification of documents in citation networks [Sen et al., 2008, Bojchevski and Günnemann, 2018], or the classification of users or posts in social networks [Shu et al., 2017].

Recently, *Graph Posterior Networks* (GPNs) have been proposed as a principled approach to UQ on graphs [Stadler et al., 2021]. The GPN model combines *Posterior Networks* (PostNets) [Charpentier et al., 2020] with the *approximate personalized propagation of neural predictions* (APPNP) node classification model [Gasteiger et al., 2018]. This combination is motivated by three axioms on how the structural information in a graph should affect the uncertainty of a model's predictions. One of those axioms states that the aleatoric entropy of nodes with conflicting neighbors should be high.

In this paper, we discuss the validity of this assumption and situations in which it does not hold. To address those situations, we propose a novel approach to uncertainty quantification on graphs based on the idea of *linear opinion pooling* (LOP) from the field of decision and risk analysis [Clemen and Winkler, 2007, Stone, 1961]. Our approach, which we refer to as *linear opinion pooled graph posterior network* (LOP-GPN), uses a mixture of Dirichlet distributions to model the uncertainty of a node's label. We demonstrate the effectiveness of our approach in a series of experiments on a variety of graph-structured datasets, showing that it outperforms existing methods in terms of both predictive accuracy and uncertainty quantification.

The remainder of this paper is organized as follows. In Section 2 we give an overview of commonly used measures for UQ. In Section 3, we review how GPNs use those measures to quantify their predictive uncertainty and then discuss the validity of this approach and describe its problems. Section 4 presents the LOP-GPNs approach that addresses the problems described in the previous section. In Section 5, we compare our approach with the original GPN model and other baseline models. Finally, Section 6 concludes the paper and outlines directions for future work.

## 2 UNCERTAINTY MEASURES

In the literature on UQ, there are different notions of what formally constitutes uncertainty. Depending on the desired properties of the uncertainty measure, different notions may be more or less suitable. We consider two ways to assess the suitability of a measure of uncertainty:

1. Its adherence to a set of axioms [Pal et al., 1993, Bronevich and Klir, 2008, Wimmer et al., 2023, Sale et al., 2023].

2. Its performance on a predictive task, such as outlier detection [Charpentier et al., 2020].

Since we focus on UQ in the node classification setting, we provide a brief overview of the most common notions of uncertainty in the context of classification tasks.

### 2.1 ENTROPY-BASED UNCERTAINTY MEASURES

One of the most common ways to represent predictive uncertainty of a $K$-class classifier is through the use of a second-order probability distribution $Q$, i.e., a distribution on the probability distributions $\theta = (\theta_1, \dots, \theta_K) \in \Delta_K$, where $\Delta_K$ is the unit $(K-1)$-simplex and $\theta_k$ denotes the probability of the $k^{th}$ class. Thus, the true distribution on the $K$ classes is considered as a random variable $\Theta \sim Q$. Given a second-order distribution $Q$, we denote its expectation by

$$\bar{\theta} := \mathbb{E}_Q[\Theta] = \int_{\Delta_K} \theta \, dQ(\theta). \quad (1)$$

The *total uncertainty* (TU) (about the outcome $Y$, i.e., the class eventually observed) can be quantified by the Shannon entropy of $\bar{\theta}$, i.e.,

$$\mathrm{TU}(Q) := H(\mathbb{E}_Q[\theta]) = -\sum_{k=1}^{K} \bar{\theta}_k \log \bar{\theta}_k. \quad (2)$$

Further, a decomposition of this uncertainty into an aleatoric and an epistemic part can be achieved on the basis of a well-known result from information theory, stating that entropy is the sum of conditional entropy and mutual information [Kendall and Gal, 2017, Depeweg et al., 2018]. This

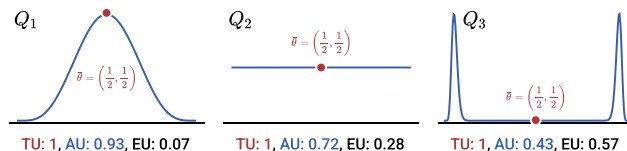

Figure 1: TU, AU, and EU for three second-order distributions on $\Delta_2$, namely, $Q_1 = \mathrm{Beta}(5, 5)$, $Q_2 = \mathcal{U}[0, 1]$, and $Q_3 = \frac{1}{2}\mathrm{Beta}(100, 10) + \frac{1}{2}\mathrm{Beta}(10, 100)$.

result suggests to quantify *aleatoric uncertainty* (AU) as conditional entropy (of the outcome $Y$ given the first-order distribution $\Theta$):

$$\mathrm{AU}(Q) := \mathbb{E}_Q[H(\Theta)] = -\int_{\Delta_K} \sum_{k=1}^{K} \theta_k \log \theta_k \, dQ(\theta). \quad (3)$$

Moreover, the *epistemic uncertainty* (EU) is then given by the difference between TU and AU, i.e.,

$$\begin{aligned} \mathrm{EU}(Q) &:= \mathrm{TU}(Q) - \mathrm{AU}(Q) \quad (4) \\ &= I(Y; \Theta) = \mathbb{E}_G[D_{\mathrm{KL}}(\Theta \| \bar{\theta})], \end{aligned}$$

where $I(\cdot; \cdot)$ denotes mutual information and $D_{\mathrm{KL}}(\cdot \| \cdot)$ the Kullback-Leibler divergence.

Figure 1 gives an intuition for the behavior of this additive decomposition of uncertainty (for $K = 2$). It shows how the AU goes down for second-order-distributions which are more concentrated around degenerate categorical distributions. As described by Wimmer et al. [2023], there are however situations in which this decomposition is less plausible. For example, $Q_2$ and $Q_3$ in Fig. 1 have the same TU, despite the fact that $Q_3$ arguably expresses more knowledge about the true data-generating distribution $\theta^*$ than $Q_2$. This raises the question, whether an additive decomposition of TU into AU and EU is reasonable at all.

Instead of expressing EU in terms of the mutual information between $Y$ and $\Theta$, Malinin and Gales [2018] and Kotelevskii et al. [2023] propose to express EU via the differential entropy of the second-order distribution $Q$, i.e.,

$$\mathrm{EU}_{\mathrm{SO}}(Q) := H(Q) = -\int_{\Delta_K} \log Q(\theta) \, dQ(\theta). \quad (5)$$

In the following, we will refer to this notion of EU as *second-order epistemic uncertainty*. Note that this notion of EU is not without controversy, as the differential entropy can be negative, which is forbidden in some axiomatic characterizations of uncertainty, which use 0 to represent a state of no uncertainty [Wimmer et al., 2023]. Apart from the entropy-based measures we just described, uncertainty is also often quantified in terms of other concentration measures, such as variance [Sale et al., 2023, Duan et al., 2024], confidence or Dirichlet pseudo-counts. We will now briefly review the latter two notions of uncertainty.

## 2.2 LEAST-CONFIDENCE- AND COUNT-BASED UNCERTAINTY MEASURES

Given a second-order distribution $Q$, a different notion of uncertainty is provided by the so-called *least-confidence* of the expected distribution $\bar{\theta}$, defined as

$$\text{LConf}(Q) \coloneqq 1 - \max_k \bar{\theta}_k \,. \tag{6}$$

Note the similarity of this measure to the TU measure in Eq. (2); $\text{LConf}(Q)$ can therefore be seen as a measure of *total* uncertainty. However, in the literature this measure is also used as a proxy for *aleatoric* uncertainty [Charpentier et al., 2020]; we will come back to this point in Section 3.

Finally, if $Q$ is described by a Dirichlet distribution $\text{Dir}(\boldsymbol{\alpha})$, where $\boldsymbol{\alpha} = (\alpha_1, \dots, \alpha_K)$ is a vector of pseudo-counts, the sum $\alpha_0 = \sum_{k=1}^K \alpha_k$ describes how concentrated $Q$ is around the expected distribution $\bar{\theta}$. Note that the concentration of $Q$ is similarly captured by its differential entropy, as described in Eq. (5), i.e., $\text{EU}_{\text{SO}}(\text{Dir}(\boldsymbol{\alpha}))$ goes down as $\alpha_0$ grows. The EU of a Dirichlet distribution $Q$ can therefore be quantified by $\text{EU}_{\text{PC}}(Q) = -\alpha_0$. We will refer to this notion of EU as *pseudo-count-based epistemic uncertainty* [Charpentier et al., 2020, Huseljic et al., 2021, Kopetzki et al., 2021].

## 3 UNCERTAINTY QUANTIFICATION

As just described, there are different ways to formalize uncertainty, and the choice of an uncertainty measure depends on the properties it is supposed to fulfill. In the context of graphs, there is an additional factor that contributes to uncertainty and needs to be formalized, too, namely the structural information. Stadler et al. [2021] propose an axiomatic approach to account for this structure-induced uncertainty, which they call *Graph Posterior Network* (GPN). As mentioned in the introduction, GPNs are essentially a combination of PostNets [Charpentier et al., 2020] and the APPNP node classification model [Gasteiger et al., 2018]. We begin with a brief review of the PostNets and GPNs, and then discuss the validity of the axioms on which the UQ estimates of GPNs are essentially based.

### 3.1 POSTERIOR NETWORKS

A PostNet is a so-called *evidential deep learning* classification model [Sensoy et al., 2018], i.e., it quantifies predictive uncertainty via a second-order distribution $Q$, which is learned via a *second-order loss function* $L_2$. A standard (first-order) loss function $L_1 : \Delta_K \times \mathcal{Y} \to \mathbb{R}$ takes a predicted first-order distribution $\hat{\theta} \in \Delta_K$ and an observed ground-truth label $y \in \mathcal{Y}$ as input (where $\mathcal{Y}$ denotes the set of classes); the *cross-entropy* (CE) loss is a common example of such a first-order loss function. Similarly, a

second-order loss $L_2$ takes a second-order distribution $Q$, i.e., a distribution over $\Delta_K$, as input, to which it again assigns a loss in light of an observed label $y \in \mathcal{Y}$. PostNet uses the so-called *uncertain cross-entropy* (UCE) loss [Biloš et al., 2019], which is defined as

$$L_2(Q, y) \coloneqq \mathbb{E}_Q\left[\text{CE}(\Theta, y)\right] \tag{7}$$
$$= -\int_{\Delta_K} \log P(y \mid \theta)\,\mathrm{d}Q(\theta)\,.$$

As shown by Bengs et al. [2022], directly minimizing a second order loss, like the UCE loss, is problematic, since the minimum is reached if $Q$ is a Dirac measure that puts all probability mass on $\theta^* = \arg\min_{\theta \in \Delta_K} \text{CE}(\theta, y)$. For all notions of EU we discussed in Section 2, the optimal $Q^*$ will therefore have no EU, i.e., $\text{EU} = 0$ (Eq. (4)) and $\text{EU}_{\text{SO}} = \text{EU}_{\text{PC}} = -\infty$ (Eq. (5)). This problem is commonly addressed by adding a *regularization term*, typically the differential entropy of $Q$, in the second-order loss function, which encourages $Q$ to be more spread out. Whether one can obtain a faithful representation of epistemic uncertainty has been generally questioned by Bengs et al. [2023]. One should therefore be cautious when interpreting the EU estimates of evidential deep learning models, such as Post-Net. We will not attempt to interpret uncertainty estimates in a quantitative manner, but rather focus on the question of whether they are qualitatively meaningful, e.g., by considering whether anomalous or noisy instances can be identified via their uncertainty.

PostNet models the second-order distribution $Q$ as a Dirichlet distribution $\text{Dir}(\alpha)$, where $\alpha = (\alpha_1, \dots, \alpha_K)$ is a vector of pseudo-counts. The predicted pseudo-counts $\alpha_k$ for a given instance $\mathbf{x}^{(i)} \in \mathcal{X}$ are defined as

$$\alpha_k = 1 + N \cdot P\left(\mathbf{z}^{(i)} \mid y^{(i)} = k\right) \cdot P\left(y^{(i)} = k\right), \tag{8}$$

where $\mathbf{z}^{(i)} = f(\mathbf{x}^{(i)}) \in \mathbb{R}^H$ is a latent neural network embedding of $\mathbf{x}^{(i)}$ and $N \in \mathbb{R}$ a so-called *certainty budget*, determining the highest attainable pseudo-count for a given instance. The class-conditional probability $P(\mathbf{z}^{(i)} \mid k)$ by a normalizing flow model for the class $k$ estimates the density of the instance. Overall, the PostNet model therefore consists of a neural network encoder model $f$ and $K$ normalizing flow models, one for each class.

### 3.2 GRAPH POSTERIOR NETWORKS

Next, we describe how the *Graph Posterior Network* (GPN) approach extends PostNet to the node classification setting for graphs. We denote a graph as $G \coloneqq (\mathcal{V}, \mathcal{E})$, where $\mathcal{V}$ is a set of $N \coloneqq |\mathcal{V}|$ nodes and $\mathcal{E} \subseteq \mathcal{V}^2$ the set of edges. The adjacency matrix of $G$ is denoted by $\mathbf{A} = (A_{i,j}) \in \{0, 1\}^{N \times N}$, where $A_{i,j} = 1$ iff $(v_i, v_j) \in \mathcal{E}$. For simplicity we also assume that $G$ is undirected, i.e., that $A$ is symmetric. For each node $v^{(i)} \in \mathcal{V}$ we have a feature vector $\mathbf{x}^{(i)} \in \mathbb{R}^D$

and a label $y^{(i)} \in \mathcal{Y}$. The goal of the node classification task is to predict the label of each node in $\mathcal{V}$, given the graph structure and the node features.

GPNs classify the nodes of a given graph by first making a prediction for each node $v^{(i)}$ solely based on its features $\mathbf{x}^{(i)}$ using a standard PostNet model, i.e., without considering the graph structure. The predicted feature-based pseudo-count vectors $\alpha^{\mathrm{ft},(i)}$ for each vertex $v^{(i)}$ are then dispersed through the graph via a *personalized page-rank* matrix $\Pi^{\mathrm{PPR}} \in \mathbb{R}^{N \times N}$ as follows:

$$\alpha^{\mathrm{agg},(i)} := \sum_{v^{(j)} \in \mathcal{V}} \Pi^{\mathrm{PPR}}_{i,j} \alpha^{\mathrm{ft},(j)} \qquad (9)$$

$$\text{where } \Pi^{\mathrm{PPR}} := \left( \varepsilon \mathbf{I} + (1 - \varepsilon) \hat{\mathbf{A}} \right)^L \qquad (10)$$

Here, $\mathbf{I}$ is the identity matrix, $\varepsilon \in (0, 1]$ the so-called *teleport probability*, and $\hat{\mathbf{A}} := \mathbf{A}\mathbf{D}^{-1}$ the normalized (random-walk) adjacency matrix, with $\mathbf{D} := \mathrm{diag}(\mathbf{A}\mathbf{1})$ being the degree matrix of $G$. For large $L$, $\Pi^{\mathrm{PPR}}$ approximates the personalized page-rank matrix of the graph via power iteration. Gasteiger et al. [2018] proposed this page-rank inspired information dispersion scheme for the node classification task, which they refer to as APPNP. The main difference between APPNP and GPN is that APPNP disperses (first-order) class probability vectors $\theta^{\mathrm{ft},(i)}$ for each node $v^{(i)}$, whereas GPN disperses pseudo-count vectors $\alpha^{\mathrm{ft},(i)}$. The latter correspond to second-order parameters, but are also in direct correspondence to zero-order (pseudo-)data. Broadly speaking, APPNP disperses first-order information, whereas GPN disperses zero-order information.

To justify this pseudo-count dispersion scheme, Stadler et al. [2021] propose three axioms on how the structural information in a graph should affect the uncertainty of a model's predictions:

**A1** A node's prediction should only depend on its own features in the absence of network effects. A node with features more different from training features should have a higher uncertainty.

**A2** All else being equal, if a node $v^{(i)}$ has a lower epistemic uncertainty than its neighbors in the absence of network effects, the neighbors' predictions should become less epistemically uncertain in the presence of network effects.

**A3** All else being equal, if a node $v^{(i)}$ has a higher aleatoric uncertainty than its neighbors in the absence of network effects, the neighbors' predictions should become more aleatorically uncertain in the presence of network effects. Further, the aleatoric uncertainty of a node in the presence of network effects should be higher if the predictions of its neighbors in the absence of network effects are more conflicting.

To show the validity of those axioms, Stadler et al. [2021] define AU to be the least-confidence (Eq. (6)) and EU as

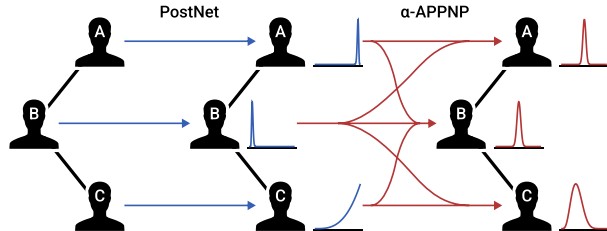

Figure 2: Illustration of how GPN aggregates the two conflicting predictions for A and B with low AU and low EU into predictions with high AU and low EU.

the sum of the pseudo-counts. Using those definitions, the validity of the axioms follows from the fact that $\alpha^{\mathrm{agg},(i)}$ is effectively a weighted average of the pseudo-counts of the (indirect) neighbors of $v^{(i)}$, with high weights for close neighbors and low weights for more distant ones.

## 3.3 VALIDITY OF THE GPN AXIOMS

The axioms that motivate the aggregation of pseudo-counts in GPNs are based on two assumptions which may not always hold, namely, *network homophily* and the *irreducibility of conflicts*.

First, *network homophily* refers to the assumption that an edge implies that the connected nodes are similar in some way; more specifically, in the context of GPNs that connected nodes should have similar second-order distributions, and thereby similar predictive uncertainties. This is a common assumption which is shared by many *graph neural network* (GNN) architectures, based on the idea of repeatedly summing or averaging the features of each node's neighbors [Kipf and Welling, 2017, Xu et al., 2018]. As already remarked by Stadler et al. [2021], non-homophilic graphs are not properly dealt with by GPNs, nor by other GNN architectures in general [Zhu et al., 2020]. Nonetheless, since edges are typically used to represent some form of similarity, the homophily assumption is often reasonable.

Second, we define the *irreducibility of conflicts* as the assumption that conflicting predictions cannot be resolved by aggregating the predictions of the conflicting nodes. Figure 2 illustrates the implications of this assumption for the binary node classification; there, without network effects, node A is very confident that its probability of belonging to the positive class is high, whereas node B is very confident that its probability of belonging to that class is low. Thus, both nodes make conflicting predictions, while both having a low AU and a low EU. Due to the homophily assumption, a consensus has to be found between the two conflicting predictions. As described in axiom **A3**, a GPN will do this by increasing the AU of the aggregated prediction, while keeping the EU low. Stadler et al. [2021] argue that this is reasonable, because such a conflict is inherently irreducible

and should therefore be reflected in the aleatoric uncertainty of the aggregated prediction.

To assess whether the irreducibility assumption is indeed reasonable, one has to clarify what *irreducibility* is actually supposed to mean. As mentioned in the introduction, irreducibility in the context of UQ refers to uncertainty that cannot be reduced by additional information, which, in a machine learning context, essentially means by sampling additional data [Hüllermeier and Waegeman, 2021]. In the context of node classification, the data points are nodes; thus given a sample graph $G_N = (\mathcal{V}_N, \mathcal{E}_N)$ with $N$ vertices, increasing the sample size corresponds to sampling a graph $G_M = (\mathcal{V}_M, \mathcal{E}_M)$ with $M > N$ nodes from an assumed underlying data-generating distribution $P_\mathcal{G}$ over all graphs $\mathcal{G}$, such that $G_N$ is a subgraph of $G_M$. The question of whether a conflict between a node $v^{(i)}$ and its neighbor $v^{(j)}$ is irreducible then becomes the question of whether the conflict persists in the limit of $M \to \infty$. Let $\mathcal{N}_M(v^{(i)})$ be the set of neighbors of $v^{(i)}$ in $G_M$. Assuming homophily, each node $v^{(\ell)}$ that is added to $\mathcal{N}_M(v^{(i)})$ should be *similar* to $v^{(i)}$ with high probability. Depending on the data-generating distribution $P_\mathcal{G}$, there are two possible scenarios:

1. The neighborhood of $v^{(i)}$ does not grow with the sample size, i.e., $\mathbb{E}[|\mathcal{N}_M(v^{(i)})|] \in \mathcal{O}(1)$ as $M \to \infty$.

2. The neighborhood of $v^{(i)}$ grows with the sample size, i.e., $\mathbb{E}[|\mathcal{N}_M(v^{(i)})|] \to \infty$ as $M \to \infty$.

In the first case, the conflict between $v^{(i)}$ and $v^{(j)}$ is indeed irreducible, as no additional data can be sampled to resolve the conflict. In this situation, axiom **A3** of GPN is reasonable, the irreducible uncertainty, i.e., AU, should increase with conflicting predictions. However, in the second case, the conflict is reducible, as the conflict will eventually be resolved by the addition of more similar nodes to the neighborhood of $v^{(i)}$, which will outweigh the conflicting node $v^{(j)}$. In this situation, the conflict resolution approach of GPN is not reasonable; AU should not go up, instead the reducible uncertainty, i.e., EU, should increase.

We argue that the second case is more common in practical node classification tasks. The Barabási-Albert model [Barabási and Albert, 1999] is a popular scale-free model, which describes the growth behavior of many real-world graphs, such as the World Wide Web, social networks, or citation networks [Albert and Barabási, 2002, Redner, 1998, Wang et al., 2008]. In this model, the expected degree of the $N$-th sampled node $v^{(i)}$ after $M - N$ additional nodes have been sampled is equal to $|\mathcal{N}_N(v^{(i)})| \cdot \sqrt{\frac{M}{N}}$. Thus, for $M \to \infty$, the expected neighborhood size of a node goes to infinity.

Examples for domains in which the neighborhood sizes do not grow with the size of a graph are molecular graphs or lattice graphs, such as 3D models or images, which can be interpreted as grids of pixels. In those domains, node-

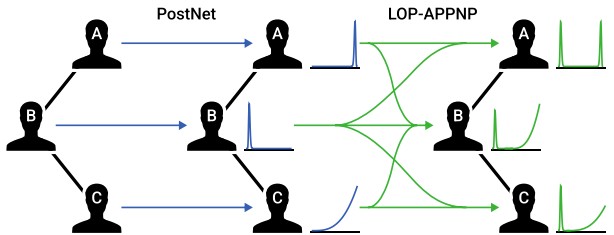

Figure 3: Illustration of how LOP-GPN preserves the AU of conflicting predictions.

level classification tasks are however less common, as one is typically interested in the classification of entire graphs, e.g., whether a given molecule is toxic or not.

To conclude, we argue that the axiomatic motivation for GPNs is oftentimes inappropriate. Therefore, we propose a different approach to UQ for node classification which does not assume the irreducibility of conflicts from axiom **A3**.

## 4 LOP-GPN

A *linear opinion pooled graph posterior network* (LOP-GPN) is a variant of the standard GPN model, which uses a mixture of Dirichlet distributions instead of a single Dirichlet distribution to model the uncertainty of a node's class in the presence of network effects. This choice is motivated by the idea that each node $v^{(i)}$ in a given node classification task can be interpreted as a decision maker, which has to assign a label $y^{(i)}$ to itself based on its own features $\mathbf{x}^{(i)}$ and the features of its neighbors. Using this interpretation, in the GPN architecture, each decision maker $v^{(i)}$ first makes a prediction based on its own features, producing a Dirichlet distribution $\mathrm{Dir}(\boldsymbol{\alpha}^{\mathrm{ft},(i)})$, and then aggregates this distribution with those produced by its direct and indirect neighbors. In the standard GPNs model this aggregate is again a Dirichlet distribution $\mathrm{Dir}(\boldsymbol{\alpha}^{\mathrm{agg},(i)})$ (see Eq. (9)).

The question of how to aggregate the decisions or opinions of a set of experts is a well-known problem in the field of decision and risk analysis [Clemen and Winkler, 2007]. One of the most common aggregation approaches in this field is the called *linear opinion pooling* (LOP) [Stone, 1961]. LOP simply aggregates distributions by taking a weighted average of their densities, resulting in a mixture of the original distributions. Combining LOP with APPNP, the aggregated probability density of a node $v^{(i)}$ is given by

$$Q^{\mathrm{agg},(i)} := \sum_{v^{(j)} \in \mathcal{V}} \Pi_{i,j}^{\mathrm{PPR}} Q^{\mathrm{ft},(j)}, \qquad (11)$$

where $Q^{\mathrm{ft},(j)}$ is the density of the Dirichlet distribution $\mathrm{Dir}(\boldsymbol{\alpha}^{\mathrm{ft},(j)})$ and $\Pi^{\mathrm{PPR}}$ as in Eq. (10). Using such mixtures, LOP-GPN does not assume the irreducibility of conflicts, i.e., unlike GPN, AU is not increased in the presence of

conflicts. This follows trivially from the fact that the AU of a LOP distribution is just the weighted linear combination of the AUs of the original distributions, i.e.,

$$
\begin{aligned}
\mathrm{AU}(Q^{\mathrm{agg},(i)}) &= \int_{\Delta_K} H(\theta)\,\mathrm{d}Q^{\mathrm{agg},(i)} \\
&= \sum_{v^{(j)}} \Pi_{i,j}^{\mathrm{PPR}} \int_{\Delta_K} H(\theta)\,\mathrm{d}Q^{\mathrm{ft},(j)} \\
&= \sum_{v^{(j)}} \Pi_{i,j}^{\mathrm{PPR}} \mathrm{AU}(Q^{\mathrm{ft},(j)}).
\end{aligned} \tag{12}
$$

Figure 3 illustrates the implications of this approach for the binary node classification. Next, we will describe how LOP-GPN is trained and how it can be implemented efficiently.

## 4.1 SECOND-ORDER LOSS

Analogous to Eq. (7), the loss of LOP-GPN is given by

$$
\mathcal{L} := \sum_{i=1}^{N} \underbrace{\mathbb{E}_{Q^{\mathrm{agg},(i)}}\left[\mathrm{CE}(\Theta^{(i)}, y^{(i)})\right] - H(Q^{\mathrm{agg},(i)})}_{\mathcal{L}^{(i)}}. \tag{13}
$$

Due to the use of Dirichlet mixtures, this loss is not directly minimizable, as there is no closed-form expression for the regularization term $H(Q^{\mathrm{agg},(i)})$. We therefore use the following bounds on the entropy of a mixture distribution instead (see Melbourne et al. [2022]):

$$
H(Q^{\mathrm{agg},(i)}) \geq \sum_{v^{(j)} \in \mathcal{V}} \Pi_{i,j}^{\mathrm{PPR}} H(Q^{\mathrm{ft},(j)}), \tag{14}
$$

$$
H(Q^{\mathrm{agg},(i)}) \leq H(\mathrm{Cat}(\Pi_i^{\mathrm{PPR}})) + \sum_{v^{(j)} \in \mathcal{V}} \Pi_{i,j}^{\mathrm{PPR}} H(Q^{\mathrm{ft},(j)}),
$$

where $\mathrm{Cat}(\Pi_i^{\mathrm{PPR}})$ is the categorical distribution described by the $i^{th}$ row vector of $\Pi^{\mathrm{PPR}}$. We use the upper bound on $H(Q^{\mathrm{agg},(i)})$ as a surrogate for $\mathrm{EU}_{\mathrm{SO}}$. Similarly, the lower entropy bound implies the following upper bound on the loss for each vertex $v^{(i)}$:

$$
\mathcal{L}^{(i)} \leq \sum_{j=1}^{N} \Pi_{i,j}^{\mathrm{PPR}} \left( \mathbb{E}_{Q^{\mathrm{ft},(i)}}\left[\mathrm{CE}(\Theta^{(i)}, y^{(i)})\right] - H(Q^{\mathrm{ft},(j)}) \right)
$$

This bound on the loss is differentiable and can therefore be minimized using standard gradient-based optimization algorithms.

## 4.2 SPARSE APPNP

Despite the similarities between GPN and LOP-GPN, the computational complexity of the APPNP-based aggregation step is significantly higher for LOP-GPN. We can express Eq. (9) as a matrix-vector multiplication, i.e.,

$$
\boldsymbol{\alpha}^{\mathrm{agg}} = \Pi^{\mathrm{PPR}} \boldsymbol{\alpha}^{\mathrm{ft}} = \hat{A}_{\varepsilon}^{L} \boldsymbol{\alpha}^{\mathrm{ft}}, \tag{15}
$$

where $\boldsymbol{\alpha}^{\mathrm{agg}}, \boldsymbol{\alpha}^{\mathrm{ft}} \in \mathbb{R}_+^{N \times K}$ are pseudo-count matrices and $\hat{A}_{\varepsilon} = \varepsilon \mathbf{I} + (1-\varepsilon)\hat{\mathbf{A}}$. Since typically $K \ll N$, it is best to evaluate this product from right to left. Then, assuming that $A$ is sparse, the complexity of this operation is $\mathcal{O}(L \cdot |\mathcal{E}| \cdot K)$.

LOP-GPN on the other hand uses the values of $\Pi^{\mathrm{PPR}}$ directly as mixture weights, i.e., the $L$-th power of $\hat{A}_{\varepsilon}$ has to be computed explicitly. For large graphs, this is computationally infeasible. To address this issue, a sparse approximation of APPNP is used to compute $\Pi^{\mathrm{PPR}}$. Between each of the $L$ sparse matrix multiplications, all probability mass below a certain threshold $\delta$ is moved back to the diagonal of the matrix. This limits the percentage of non-zero entries to $(N\delta)^{-1}$ and makes it possible to apply LOP-GPN even to large graphs. In the experimental evaluation, sparsification was used for all datasets except CoraML and CiteSeer.

## 5 EVALUATION

We evaluate LOP-GPN in two ways: First, we use *accuracy-rejection curves* (ARCs) to compare the quality of different predictive uncertainty measures of LOP-GPN to those of GPN on a set of standard node classification benchmarks. Second, we compare the *out-of-distribution* (OOD) detection performance of our model against a set of node classification models.

## 5.1 EXPERIMENTAL SETUP

Due to the similarity between LOP-GPN and GPN, we base our experiments on those of Stadler et al. [2021], i.e., we use the same dataset splits and hyperparameters and build upon their reference implementation.[1]

**Datasets** We use the following node classification benchmarks: Three citation network datasets, namely, **CoraML**, **CiteSeer** and **PubMed** [McCallum et al., 2000, Giles et al., 1998, Getoor, 2005, Sen et al., 2008, Namata et al., 2012], two co-purchase datasets, namely, **Amazon Photos** and **Amazon Computers** [McAuley et al., 2015] and the large-scale **OGBN Arxiv** dataset with about 170k nodes and over 2.3 million edges [Hu et al., 2020]. Since OGBN Arxiv is presplit into train, validation and test sets, we use the provided splits. The results of the other datasets are obtained by averaging over 10 dataset splits with train/val/test sizes of 5%/15%/80%. Note that all six datasets represent either citation networks or co-purchase networks; the assumptions of unbounded growth of neighborhood sizes and network homophily (see Section 3.3) are therefore reasonable and the use of LOP-GPN well-motivated.

---

[1]Implementation available at https://github.com/Cortys/gpn-extensions

**Models** We compare **LOP-GPN** against the following baseline models: Two variants of **GPN** [Stadler et al., 2021], **APPNP** [Gasteiger et al., 2018], **Matern-GGP** [Borovitskiy et al., 2021] and **GKDE** [Zhao et al., 2020].

As described in Section 3.2, **APPNP** directly disperses class probabilities, i.e., it is a first-order method that cannot (meaningfully) distinguish between AU and EU; the entropy of its first-order predictions is therefore used as an estimate of total uncertainty.

**Matern-GGP** [Bojchevski and Günnemann, 2018] is a Gaussian process model using the so-called *graph Matérn kernel*. For each vertex, it predicts a posterior second-order multivariate Gaussian $\Theta \sim \mathcal{N}(\bar{\theta}, \Sigma)$ with $\bar{\theta} \in \Delta_K$ and $\Sigma$ a diagonal covariance matrix. This implies that some probability mass will lie outside of $\Delta_K$; thus, the additive entropy-based uncertainty decomposition described in Section 2.1 is not well-defined for this distribution. We therefore do not use the additive entropy decomposition for Matern-GGP and instead use $H(\bar{\theta})$ as a proxy for AU and the trace $\mathrm{Tr}(\Sigma)$ as a proxy for EU. Note that this definition of AU coincides with the definition of TU in the additive decomposition (Eq. (2)). Second, note that $\mathrm{Tr}(\Sigma)$ is a monotonic transformation of the differential entropy of the second-order Gaussian $\mathcal{N}(\bar{\theta}, \Sigma)$ and thereby closely related to $\mathrm{EU_{SO}}$ (Eq. (5)).[2] Last, we note that the Matern-GGP model was not applied to the large-scale OGBN Arxiv dataset due to memory constraints.

The *Graph-based Kernel Dirichlet distribution Estimation* **GKDE** model [Zhao et al., 2020], is a parameter-free method that estimates a Dirichlet distribution for each node based on the features of its neighbors.

The two evaluated variants of **GPN** are *GPN (rw)* and *GPN (sym)*. GPN (rw) uses APPNP with random-walk normalization, i.e., with $\hat{\mathbf{A}} = \mathbf{A}\mathbf{D}^{-1}$. GPN (sym) uses symmetric normalization, i.e., with $\hat{\mathbf{A}} = \mathbf{D}^{-1/2}\mathbf{A}\mathbf{D}^{-1/2}$ (see Kipf and Welling [2017]). We evaluate both types of normalization because Stadler et al. [2021] used symmetric normalization in their experiments, while LOP-GPN requires random-walk normalization to ensure that valid mixture densities are produced. To show that any observed differences between LOP-GPN and GPN are due to the use of LOP and not due to the use of random-walk normalization, we compare LOP-GPN against both variants.

## 5.2 ACCURACY-REJECTION CURVES

*Accuracy-rejection curve*s (ARCs) are produced by discarding the predictions for instances where the predictor exhibits the highest uncertainty, and then calculating the accuracy for the remaining subset. For an uncertainty measure which captures predictive uncertainty well, the accuracy should monotonically increase with the rejection rate. We use the following uncertainty measures to produce ARCs: Entropy-based TU, AU and EU (Eqs. (2) to (4)), second-order epistemic uncertainty (Eq. (5)), and pseudo-count-based epistemic uncertainty. Figure 4 show the ARCs for the different uncertainty measures for the six evaluated datasets. First, note that LOP-GPN consistently outperforms or at least matches GPN for almost all rejection rates and all datasets. This indicates that dropping the assumption of the *irreducibility of conflicts* made by GPN is beneficial in practice. The only exception to this are the curves for the mutual information-based EU on the CiteSeer, Amazon Photos and Amazon Computers datasets; here, the accuracies of LOP-GPN drop-off at high rejection rates. As previously illustrated in Fig. 1, the mutual information-based EU can be smaller for the uninformed uniform second-order distribution than for a (more-informed) bimodal Dirichlet mixture distribution. Analogous to that example, we hypothesize that some of the uninformed mixture distributions produced by LOP-GPN are incorrectly assigned low EU values, leading to the observed drop-off in accuracy for the nodes with the lowest EU estimates. The (arguably) more reasonable pseudo-count- and differential entropy-based EU measures do not exhibit this behavior. There, the accuracies go up for increasing rejection rates; this is a sign that these measures capture predictive uncertainty in a meaningful way.

## 5.3 OUT-OF-DISTRIBUTION DETECTION

Next, we evaluate the performance of LOP-GPN and the other models on *out-of-distribution* (OOD) detection tasks. Similar to Stadler et al. [2021], we use three different OOD detection tasks. First, we evaluate the models' ability to detect nodes belonging to classes that were not present in the training set. Second, we randomly drop features from some nodes with probability $0.5$ and evaluate the models' ability to detect those nodes as outliers. Third, we add Gaussian noise to the features of some nodes, which should then be detected as outliers. For all detection tasks, we use the entropy-based TU, AU and EU measures, as well as $\mathrm{EU_{PC}}$ and $\mathrm{EU_{SO}}$, as criteria. We use the *Area Under Receiving Operator Characteristics Curve* (AUC-ROC) to measure the performance of the models on the OOD nodes; the performance on the *in-distribution* (ID) training data is measured via the accuracy.

Table 1 shows the OOD detection performance and ID accuracies. Overall, LOP-GPN performs very well on the feature dropout and Gaussian noise tasks, outperforming the other models most often. Looking at the ID accuracies in the Gaussian noise setting, LOP-GPN is uniquely able to achieve high accuracies on all but the OGBN Arxiv dataset. On the leave-out-classes detection task, LOP-GPN performs slightly worse but overall still similarly to the other models.

---

[2] The differential entropy of the multinomial Gaussian $\mathcal{N}(\bar{\theta}, \Sigma)$ is $\frac{1}{2}\ln\left((2\pi e)^k \det \Sigma\right) = \frac{1}{2}\mathrm{Tr}(\ln \Sigma) + c \geq \frac{1}{2}\ln \mathrm{Tr}(\Sigma) + c$ with the constant $c = \frac{k}{2}\ln(2\pi e)$.

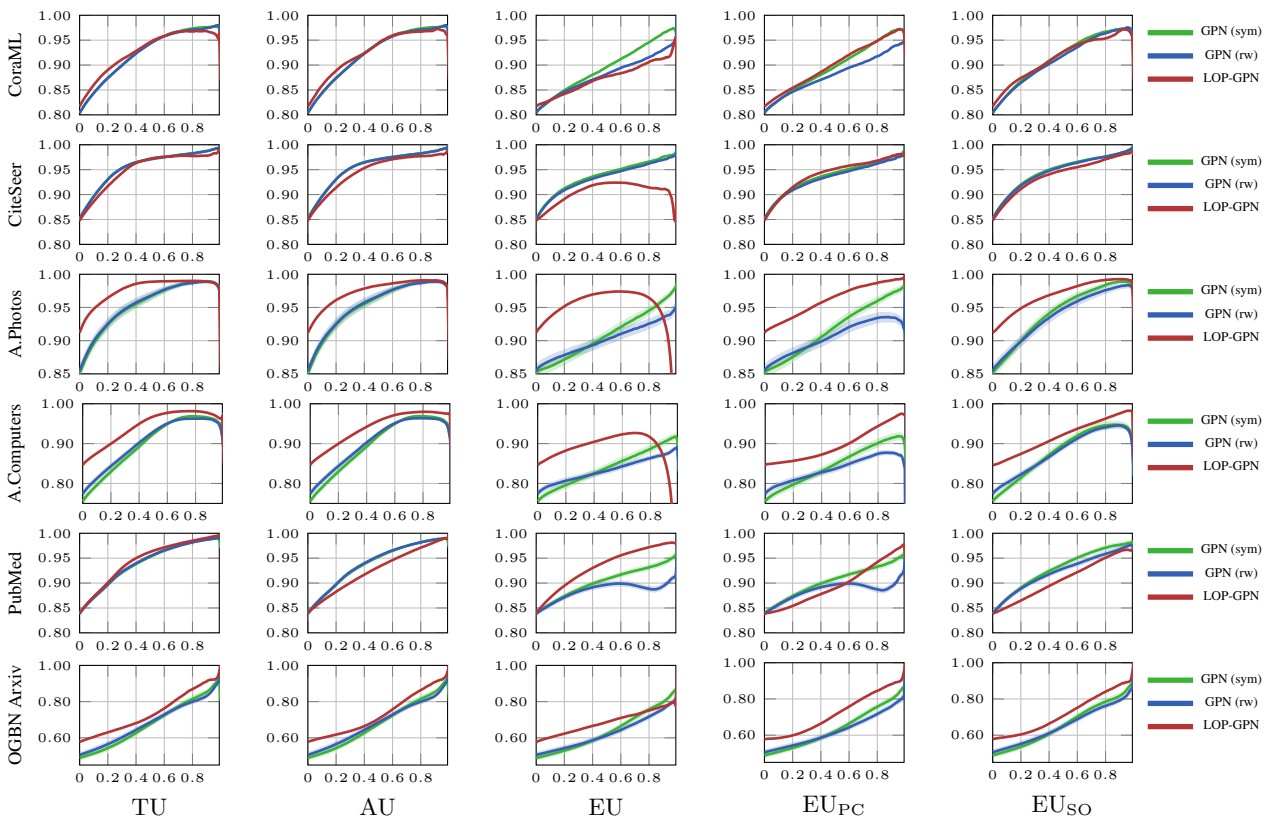

Figure 4: Accuracy-rejection curve for different uncertainty measures. The x-axis represents the fraction of rejected test instances; the y-axis represents the test accuracy for a given rejection rate. The (small) shaded areas behind the curves represent the estimate's standard error.

Table 1: OOD detection performance of OOD vs ID vertices and ID accuracies for three OOD scenarios.

| | | Leave-out Classes | | | | | | $\mathbf{x}^{(v)} \sim \text{Ber}(0.5)$ | | | | | | $\mathbf{x}^{(v)} \sim \mathcal{N}(0,1)$ | | | | | |
| | | ID | | OOD-AUC-ROC | | | | ID | | OOD-AUC-ROC | | | | ID | | OOD-AUC-ROC | | | |
| | | Acc | TU | AU | EU | $EU_{PC}$ | $EU_{SO}$ | Acc | TU | AU | EU | $EU_{PC}$ | $EU_{SO}$ | Acc | TU | AU | EU | $EU_{PC}$ | $EU_{SO}$ |
|---|---|---|---|---|---|---|---|---|---|---|---|---|---|---|---|---|---|---|---|
| CoraML | APPNP | **90.44** | **87.45** | - | - | - | - | **84.14** | **65.02** | - | - | - | - | 43.62 | 13.26 | - | - | - | - |
| | Matern-GGP | 85.49 | 82.31 | 82.31 | - | - | 82.18 | 77.15 | 49.71 | 49.71 | - | - | 49.81 | 77.15 | 49.71 | 49.71 | - | - | 49.81 |
| | GKDE | 83.01 | 77.21 | 35.35 | 69.16 | 74.00 | 76.46 | 71.96 | 48.71 | 50.76 | 49.19 | 48.45 | 48.68 | 71.96 | 48.71 | 50.76 | 49.19 | 48.45 | 48.68 |
| | GPN (sym) | 89.36 | 85.51 | 85.49 | **86.23** | **87.11** | **89.15** | 80.36 | 54.44 | 54.44 | 51.56 | 54.17 | 87.19 | 17.86 | **96.59** | **96.80** | 70.89 | **70.62** | 75.66 |
| | GPN (rw) | 89.30 | 85.19 | 85.17 | 82.52 | 83.11 | 87.60 | 80.51 | 55.09 | 55.09 | 51.88 | 54.32 | **87.67** | 17.86 | 92.65 | 93.22 | 63.40 | 63.21 | 66.91 |
| | LOP-GPN | 89.34 | 85.67 | **88.51** | 45.18 | 84.72 | 79.26 | 81.29 | 62.20 | **59.69** | **64.31** | 54.40 | 56.74 | **81.74** | 69.50 | 61.06 | **82.08** | 60.15 | **83.69** |
| CiteSeer | APPNP | **87.75** | **85.80** | - | - | - | - | **85.65** | 70.25 | - | - | - | - | 72.72 | 23.76 | - | - | - | - |
| | Matern-GGP | 49.05 | 78.56 | 78.56 | - | - | 78.59 | 50.02 | 50.56 | 50.56 | - | - | 50.55 | 50.02 | 50.56 | 50.56 | - | - | 50.55 |
| | GKDE | 73.73 | 77.89 | 37.03 | 65.08 | **81.30** | 79.44 | 65.79 | 50.14 | 51.33 | 48.80 | 50.17 | 50.04 | 65.79 | 50.14 | 51.33 | 48.80 | 50.17 | 50.04 |
| | GPN (sym) | 87.23 | 81.53 | **81.52** | **76.78** | 76.76 | **80.62** | 84.82 | 62.58 | 62.58 | 53.65 | **60.16** | **91.73** | 17.45 | **93.00** | **93.22** | 80.32 | **79.91** | 86.34 |
| | GPN (rw) | 87.12 | 81.23 | 81.23 | 74.29 | 74.26 | 79.61 | 84.78 | 63.16 | 63.16 | 53.78 | 60.15 | 91.59 | 17.45 | 91.28 | 91.68 | 72.38 | 72.00 | 78.25 |
| | LOP-GPN | 87.16 | 82.19 | 80.57 | 73.09 | 75.48 | 73.12 | 84.31 | **78.74** | **78.43** | **68.54** | 58.12 | 68.98 | **84.41** | 81.65 | 76.92 | **81.26** | 59.20 | **86.81** |
| Amazon Photos | APPNP | **94.99** | 75.12 | - | - | - | - | **91.83** | 63.55 | - | - | - | - | 40.44 | 13.17 | - | - | - | - |
| | Matern-GGP | 88.57 | 82.12 | 82.12 | - | - | 82.68 | 86.04 | 49.62 | 49.62 | - | - | 49.62 | 86.04 | 49.62 | 49.62 | - | - | 49.62 |
| | GKDE | 85.45 | 70.20 | 55.45 | 61.23 | 60.80 | 66.83 | 76.19 | 49.07 | 50.74 | 48.76 | 48.87 | 48.95 | 76.19 | 49.07 | 50.74 | 48.76 | 48.87 | 48.95 |
| | GPN (sym) | 91.49 | 76.29 | 76.29 | **86.54** | **87.50** | **86.05** | 84.79 | 54.55 | 54.55 | 49.87 | 54.29 | 80.17 | 12.63 | 89.43 | **91.07** | 60.06 | 59.86 | 63.56 |
| | GPN (rw) | 91.76 | 76.22 | 76.22 | 77.93 | 78.71 | 81.72 | 85.35 | 55.38 | 55.38 | 49.92 | 53.33 | 74.95 | 12.63 | 91.70 | 81.36 | 56.12 | 55.99 | 58.16 |
| | LOP-GPN | 94.00 | **86.50** | **83.88** | 80.02 | 76.63 | 83.87 | 91.10 | **79.21** | 70.67 | 78.59 | 53.39 | 61.00 | **91.18** | **91.76** | 87.29 | **89.36** | **61.15** | **95.74** |
| Amazon Computers | APPNP | 87.99 | 79.32 | - | - | - | - | 81.11 | 64.62 | - | - | - | - | 42.81 | 15.58 | - | - | - | - |
| | Matern-GGP | 86.74 | 82.20 | 82.20 | - | - | 81.94 | 81.00 | 50.02 | 50.02 | - | - | 50.00 | 81.00 | 50.02 | 50.02 | - | - | 50.00 |
| | GKDE | 71.26 | 76.38 | 70.52 | 74.46 | 74.37 | 76.20 | 64.01 | 49.92 | 49.81 | 50.03 | 50.09 | 49.99 | 64.01 | 49.92 | 49.81 | 50.03 | 50.09 | 49.99 |
| | GPN (sym) | 82.17 | 79.17 | 79.17 | **76.65** | **81.01** | 83.77 | 75.71 | 54.67 | 54.67 | 51.14 | **56.46** | 85.57 | 16.39 | 88.69 | 89.62 | 62.32 | 62.12 | 65.90 |
| | GPN (rw) | 84.45 | 79.45 | 79.45 | 72.80 | 76.01 | 81.79 | 79.47 | 55.44 | 55.44 | 51.15 | 55.25 | **86.11** | 16.39 | 79.77 | 80.82 | 58.38 | 58.25 | 60.57 |
| | LOP-GPN | **90.28** | **85.00** | **88.08** | 68.40 | 77.98 | **83.80** | **84.35** | **79.78** | **72.26** | **75.86** | 51.59 | 61.91 | **84.49** | **94.76** | **90.93** | **88.33** | **65.30** | **98.15** |
| PubMed | APPNP | **94.59** | 69.38 | - | - | - | - | **86.73** | 66.90 | - | - | - | - | 59.59 | 10.95 | - | - | - | - |
| | Matern-GGP | 90.05 | 63.72 | 63.72 | - | - | 63.72 | 78.62 | 50.14 | 50.14 | - | - | 50.13 | 78.62 | 50.14 | 50.14 | - | - | 50.13 |
| | GKDE | 87.93 | **71.51** | 39.69 | 65.30 | 69.74 | 71.00 | 76.96 | 49.94 | 50.16 | 49.79 | 49.87 | 49.93 | 76.96 | 49.94 | 50.16 | 49.79 | 49.87 | 49.93 |
| | GPN (sym) | 93.76 | 69.87 | **69.85** | **72.95** | **72.93** | **73.62** | 83.89 | 58.49 | 58.49 | 51.12 | **60.66** | 81.42 | 30.29 | **95.74** | **96.41** | 70.52 | **70.29** | 76.44 |
| | GPN (rw) | 94.06 | 69.52 | 69.50 | 67.69 | 67.66 | 72.20 | 83.94 | 60.36 | 60.36 | 51.56 | 59.66 | 81.13 | 30.29 | 91.19 | 92.56 | 66.13 | 65.92 | 70.80 |
| | LOP-GPN | 93.23 | 68.96 | 69.10 | 66.28 | 64.87 | 64.30 | 83.52 | 70.46 | 67.93 | 65.49 | 55.83 | 60.86 | **83.99** | 82.71 | 75.93 | 80.37 | 67.74 | **91.84** |
| OGBN Arxiv | APPNP | **73.33** | 64.47 | - | - | - | - | **68.22** | 66.97 | - | - | - | - | **66.53** | 42.63 | - | - | - | - |
| | GKDE | 60.04 | **68.69** | 66.97 | 66.70 | **67.46** | 68.14 | 56.91 | 49.52 | 49.55 | 49.48 | 49.47 | 49.46 | 56.91 | 49.52 | 49.55 | 49.48 | 49.47 | 49.46 |
| | GPN (sym) | 55.36 | 67.08 | 67.08 | **66.92** | 66.91 | **68.79** | 50.35 | 51.15 | 51.15 | 50.95 | 53.41 | 97.12 | 4.35 | 77.08 | **77.45** | 67.14 | 66.92 | 69.93 |
| | GPN (rw) | 56.87 | 67.99 | 67.99 | 63.37 | 63.36 | 66.72 | 50.93 | 51.56 | 51.56 | 51.26 | 52.94 | **97.25** | 4.35 | 72.27 | 72.76 | 60.50 | 60.35 | 62.39 |
| | LOP-GPN | 63.37 | 66.77 | **68.12** | 54.58 | 66.28 | 67.44 | 57.85 | **70.58** | **68.03** | **64.08** | **63.98** | 72.86 | 57.49 | **82.46** | 73.03 | **87.08** | **68.39** | **92.39** |

To summarize our experimental results, LOP-GPN was able to achieve strong classification accuracies and meaningful uncertainty estimates, as shown in Fig. 4 and Table 1. This supports our hypothesis that the irreducibility of conflicts assumption made by GPN is not always justified in real-world node classification tasks.

# 6 CONCLUSION

In this paper, we proposed a new approach to uncertainty quantification in (semi-supervised) node classification, which is able to represent both aleatoric and epistemic uncertainty. Broadly speaking, while existing methods realize information dispersion on the level of the data (or pseudo-counts) or the level of aleatoric uncertainty (averaging first-order distributions), our approach makes use of the graph structure to combine information directly on the epistemic level. To this end, we refer to the established principle of linear opinion pooling and represent epistemic uncertainty in terms of mixtures of Dirichlet distributions. First experiments on a variety of graph-structured datasets are promising and show the effectiveness of our approach, also compared to state-of-the-art methods used as baselines.

In future work, we plan to study the problem of uncertainty propagation on graphs in more depths and to compare different approaches in a more systematic way. Intuitively, an optimal approach should find a good compromise between combining information on the aleatoric and the epistemic level, respectively. However, for now, it is not at all clear how such an approach could be realized.

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
