# OpenReview forum: "Linear Opinion Pooling for Uncertainty Quantification on Graphs"
_auai.org/UAI/2024/Conference — UAI 2024 poster_

### Official Review · Reviewer_JeuK · 2024-03-20

**Q2-1 Originality-Novelty:** 3
**Q2-2 Correctness-Technical Quality:** 3
**Q2-5 Clarity Of Writing:** 4

**Q1 Summary And Contributions:**

This manuscript tackles the challenging problem of uncertainty quantification for graph-structured data, specifically focusing on predictive uncertainty in node classification. The article addresses the distinction between aleatoric and epistemic uncertainty and proposes a novel approach that utilizes mixtures of Dirichlet distributions and linear opinion pooling to represent and propagate uncertainty within the graph.

**Q2-3 Extent To Which Claims Are Supported By Evidence:**

4: Excellent: all claims are supported by very convincing evidence (in the form of comprehensive experimental evaluation, rigorous mathematical proofs, detailed (pseudo-)code, precise references, well-motivated and realistic assumptions) and the authors deliver what they promise.

**Q2-4 Reproducibility:**

3: Good: key resources (e.g. proofs, code, data) are available and key details (e.g. proofs, experimental setup) are sufficiently well-described for competent researchers to confidently reproduce the main results.

**Q3 Main Strengths:**

The manuscript is clear and well-written. The theoretical section is easy to follow even for someone (like myself) with limited expertise in this area, and the copious citations suggest a strong engagement with the existing literature. The empirical results are compelling.

**Q4 Main Weakness:**

I enjoyed this paper and have no substantive revisions to suggest. One question that I wasn't entirely sure about – it was not clear to me whether the uncertainty quantification of LOP-GPN was modular with respect to the node classifier. In other words, could we combine this method with any node classification scheme to get valid results or is it tied to some particular class of node classifiers?

**Q5 Detailed Comments To The Authors:**

As noted above, I have no major comments. One minor comment – it would be helpful to put the best performing model in boldface in the tables.

**Q9 Complying With Reviewing Instructions:**

Yes

---

> ### Author Rebuttal · Authors · 2024-04-03
>
> First of all, thank you for the positive feedback and the interesting question regarding the modularity of the proposed method!
>
> Whether one can/should consider LOP-GPN as "modular" depends on what one considers to be a "node classifier" and which aspects of LOP-GPN one considers to be the "uncertainty quantification" (UQ) component.
>
> LOP-GPN essentially consists of two components:
> 1. A PostNet [3] density estimation model, which processes each node independently and predicts second-order Dirichlet distributions for each node.
> 2. A fixed graph dispersion scheme, which produces a Dirichlet mixture distribution for each node.
>
> Both components are involved in the UQ process of the LOP-GPN model.
> The density estimator is used to produce structure-independent pseudo-counts; it thereby also produces structure-independent uncertainty estimates for each node.
> The graph dispersion scheme then incorporates structural information into the uncertainty estimates from the first step.
>
> Since all components of LOP-GPN are involved in the uncertainty quantification process, it is therefore not really possible to "swap out" the node classifier part, since UQ and node classification are tightly interwoven.
> From this perspective, LOP-GPN is not modular with respect to the node classifier.
>
> However, it is important to note that the main novelty of LOP-GPN lies in how it disperses the structure-unaware uncertainty estimates using Dirichlet mixture distributions.
> If one only considers this Dirichlet mixture approach as the "uncertainty quantification of LOP-GPN", the approach could indeed be called modular with respect to the node classifier.
> More specifically, instead of using the PostNet approach for structure-independent node classification, any other structure-unaware classifier producing second-order (Dirichlet) distributions could be used [1,2].
>
> While evaluating such alternative combinations could be interesting, it is important to note that PostNet was proposed to address problems with the two cited previous approaches; whether this combination would produce better results in the node classification setting is therefore at least questionable.
>
> We hope that you find these additional explanations helpful.
>
> Last, we note that we will update the tables and put the best results in boldface.
>
> Thank you, once again, for your feedback!
>
> ---
>
> [1] Sensoy, M.; Kaplan, L.; Kandemir, M. Evidential Deep Learning to Quantify Classification Uncertainty. In _Proceedings of the 32nd International Conference on Neural Information Processing Systems_; NIPS’18; Curran Associates Inc.: Red Hook, NY, USA, 2018; pp 3183–3193.
>
> [2] Malinin, A.; Gales, M. Predictive Uncertainty Estimation via Prior Networks. In _Proceedings of the 32nd International Conference on Neural Information Processing Systems_; NIPS’18; Curran Associates Inc.: Red Hook, NY, USA, 2018; pp 7047–7058.
>
> [3] Charpentier, B.; Zügner, D.; Günnemann, S. Posterior Network: Uncertainty Estimation without OOD Samples via Density-Based Pseudo-Counts. In _Proceedings of the 34th International Conference on Neural Information Processing Systems_; NIPS’20; Curran Associates Inc.: Red Hook, NY, USA, 2020; pp 1356–1367.

---

### Official Review · Reviewer_BPUb · 2024-03-22

**Q2-1 Originality-Novelty:** 3
**Q2-2 Correctness-Technical Quality:** 3
**Q2-5 Clarity Of Writing:** 3

**Q1 Summary And Contributions:**

This paper explores uncertainty quantification on graph data, particularly, in node classification tasks. The authors proposed a novel method to represent epistemic uncertainty and utilise the graph topology information via linear opinion pooling. Then, they demonstrate the effectiveness of their method on a range of benchmark datasets.

**Q2-3 Extent To Which Claims Are Supported By Evidence:**

3: Good: the main claims are supported by convincing evidence (in the form of adequate experimental evaluation, proofs, (pseudo-)code, references, assumptions).

**Q2-4 Reproducibility:**

3: Good: key resources (e.g. proofs, code, data) are available and key details (e.g. proofs, experimental setup) are sufficiently well-described for competent researchers to confidently reproduce the main results.

**Q3 Main Strengths:**

The general presentation of this paper is good and easy to follow. I am not an expert in this particular field but I think in general the methodology make sense to me and the results seems convincing.

**Q4 Main Weakness:**

I think the paper can be better structured and provide some more technical details in section 2 and 3. I will put some of my questions in the Q5 section.

**Q5 Detailed Comments To The Authors:**

The replacement of single Dirichlet distribution with a mixture of decision nodes, while each node is a single Dirichlet distribution seems reasonable to me. With the LOP-GPN formulation, it seems like to me the aggregation of information (message) is localised and linear, which means the EU is not transmitted outside the local community. Would this not be an issue? Could you please comments on this and if possible briefly discuss this concern?

**Q9 Complying With Reviewing Instructions:**

Yes

---

> ### Author Rebuttal · Authors · 2024-04-03
>
> Thank you for your constructive feedback and the important question you raised!
>
> In Sections 2 and 3 we focused on providing the context needed to understand the motivation and definition of our proposed LOP-GPN approach. Some technical details, especially about GPNs, were therefore left out. We are, of course, open to providing more details on the related work. Let us know if there are any specific aspects for which you would like to have additional information.
>
> Regarding your question on the localized aggregation/dispersion of information: As you pointed out, the dispersion scheme used by (LOP-)GPN is indeed linear and local.
> Whether this is an issue or not depends on two factors:
> 1. _Do nodes that are far apart have a relevant influence on each-other?_
>    Due to the fact that edges are often used to represent node similarity, an increasing distance between nodes corresponds to increasing dissimilarity. Both, GPN and LOP-GPN, make this so-called _homophily_ assumption. Under this assumption it is, generally, not useful to assign a large weight $\Pi^{\mathrm{PPR}}_{i,j}$ to node pairs $v^{(i)}, v^{(j)}$ that are far apart. Limiting propagation to a local community is therefore a reasonable decision in many situations.
> 2. _How restrictive is the chosen locality radius?_
>    The size of the local neighborhoods depends on the parameter $L$ (see eq. 10). If $L$ is larger than or equal to the diameter of the graph, i.e., the largest distance between any two nodes, all nodes can influence each other and the local dispersion scheme effectively turns into a global one.
>    Should the sizes of the neighborhoods be too restrictive/local for a specific problem, one can therefore easily increase $L$ such that all relevant indirect neighbors of a node will be considered. Increasing $L$ does, of course, come at the cost of increasing the computational complexity of the model. Additionally, as noted in the first point, making $L$ too large could lead to dissimilar nodes influencing each other, even though they should not.
>
> To summarize, there is no universally "correct" degree of locality for a information dispersion scheme in graphs; this choice is problem-specific and should be treated as a tunable hyperparameter of the model.
> In our experiments, we chose $L = 10$. This value was also used in the original GPN paper and empirically provided good results on the used benchmark datasets.
> Given the datasets, this choice of $L$ results in a (nearly) global dispersion scheme, since most of the used datasets have diameters of 10 or slightly more than 10 (e.g., AmazonComputers has a diameter of 10, AmazonPhotos of 11 and CoraML of 17).
>
> We hope that this explanation is helpful and that it answers your question. Thank you for your feedback, once again.

---

### Official Review · Reviewer_A7qh · 2024-03-23

**Q2-1 Originality-Novelty:** 3
**Q2-2 Correctness-Technical Quality:** 3
**Q2-5 Clarity Of Writing:** 4

**Q1 Summary And Contributions:**

In this paper the authors propose a new approach to uncertainty quantification in semi-supervised node classification. Their approach combines aleatoric and epistemic uncertainty. In Section 3, they convincingly argue what the problem is with the standard Graph Posterior Network (GPN) approach, and identify conditions under which their axioms are not always defensible.

Instead, they propose in Section 4 a linear opinion pooled GPN, and study its properties.

**Q2-3 Extent To Which Claims Are Supported By Evidence:**

3: Good: the main claims are supported by convincing evidence (in the form of adequate experimental evaluation, proofs, (pseudo-)code, references, assumptions).

**Q2-4 Reproducibility:**

2: Fair: key resources (e.g. proofs, code, data) are unavailable but key details (e.g. proof sketches, experimental setup) are sufficiently well-described for an expert to confidently reproduce the main results.

**Q3 Main Strengths:**

The topic of this paper is interesting and relevant to the UAI conference.
Its relevance is convincingly argued by the authors: they carefully discuss in Section 3 when the current standard approach fails. I like this section.

In my opinion, this paper is very well written. The authors took great care in carefully explaining the problem and their proposed solutions, using a clear language.

**Q4 Main Weakness:**

I identify two weaknesses. As I think it is a good paper, the weaknesses are not too severe in my opinion.

1. Section 4 could do with some more explaining, such as discussion of the properties, and perhaps computational efficiency.

2. The authors don't provide code for their experiments. As their experiments are closely based on a different paper, they could likely be retrieved from there, but it would improve the reproducibility if the authors had provided some code.

**Q5 Detailed Comments To The Authors:**

Here are my detailed comments.

Page 1, right column: "adress" should be "address".

Page 3, left column: "classificatoin" should be "classification".

Page 4, Axiom A1: Even though it is a quote from [Stadler et al. 2021], I don't find the second sentence clear ("A node with feature more different from training features ...").

Page 4, right column: "assuption" should be "assumption".

Page 6, right column, final paragraph before Section 5: "compuationally" should be "computationally".

Page 7, left column, final sentence before Section 5.2: "againt" should be "against".

Page 7, right column, final sentence: "It" should be "it", so without a capital.

**Q9 Complying With Reviewing Instructions:**

Yes

---

> ### Author Rebuttal · Authors · 2024-04-03
>
> We thank the reviewer for their thorough review and useful suggestions!
>
> Regarding the discussion of LOP-GPNs in Section 4: Since our approach shares many similarities with the standard GPN model, we focused mainly on the differences between GPN and LOP-GPN. As alluded to in Sections 4.1 and 4.2, LOP-GPN is indeed more computationally expensive than GPN, necessitating the use of approximations for the loss and the graph diffusion mechanism.
>
> In a connected graph with $N$ vertices, the Dirichlet mixtures $\mathcal{Q}^{\mathrm{agg},(i)}$ at each vertex $v^{(i)}$ can consist of up to $N$ components, one for each vertex; in LOP-GPN $\mathcal{Q}^{\mathrm{agg},(i)}$ is therefore parameterized by $N \cdot K$ pseudo-counts, with $K$ being the number of classes. The single Dirichlet distributions of GPN, on the other hand, only have $K$ parameters. As described in Section 4.2, this difference increases the computational complexity of LOP-GPN. To keep the runtime tractable we introduce the pruning threshold $\delta \in [0,1]$ which acts as a lower bound on the weight of each mixture component and thereby limits the number of mixture components in $\mathcal{Q}^{\mathrm{agg},(i)}$ to $\delta^{-1}$.
>
> We could indeed expand Section 4 by a more detailed discussion of this approximation and its implications. Thank you for this suggestion!
>
> Regarding the code for the experiments: Our implementation indeed builds upon the reference implementation for GPN. To improve reproducibility, we will also publish our code; it was not included in the initial submission because we would like to clean it up and add some documentation beforehand.
>
> Last, thank you for the list of typographical mistakes! Those will, of course, also be fixed.
>
> We appreciate your feedback and hope that this addresses your suggestions. Let us know if you have any other questions or remarks.

---

### Official Review · Reviewer_6J7q · 2024-03-24

**Q2-1 Originality-Novelty:** 2
**Q2-2 Correctness-Technical Quality:** 3
**Q2-5 Clarity Of Writing:** 3

**Q1 Summary And Contributions:**

The paper proposes a novel approach to uncertainty quantification on graphs vased on the idea of linear opinion pooling. The effectiveness of the approach has been validated in a series of experiments on a number of graph-structured datasets.

**Q2-3 Extent To Which Claims Are Supported By Evidence:**

3: Good: the main claims are supported by convincing evidence (in the form of adequate experimental evaluation, proofs, (pseudo-)code, references, assumptions).

**Q2-4 Reproducibility:**

3: Good: key resources (e.g. proofs, code, data) are available and key details (e.g. proofs, experimental setup) are sufficiently well-described for competent researchers to confidently reproduce the main results.

**Q3 Main Strengths:**

- Clear formalization
- The problem addressed in of interest in pratical applications

**Q4 Main Weakness:**

- The work found deeply on GPNs

**Q5 Detailed Comments To The Authors:**

- Could you please related your work with existing approaches in the literature (in particular those reported in the experimental analysis)?

**Q9 Complying With Reviewing Instructions:**

Yes

---

> ### Author Rebuttal · Authors · 2024-04-06
>
> Thank you for your constructive feedback!
>
> As pointed out, our work is directly based on GPNs, which is a state-of-the-art approach for graph uncertainty quantification. We agree that replacing the second-order Dirichlet distributions in GPN by mixture distributions does, _by itself_, not result in a fundamentally novel model - this is also reflected in the name, _LOP-GPN_.
>
> Our main contribution lies in the theoretical motivation and the empirical evidence we provide for this model. To summarize, the main contribution of our paper is threefold:
>
> 1. To our knowledge, this is the first work describing the influence of the problem domain on the appropriateness of graph uncertainty quantification approaches.
> 2. We argue that the notion of epistemic uncertainty used by GPNs is often not appropriate for node classification tasks (Section 3.3). More specifically, we show that the _irreducibility of conflicts_ assumption made by GPNs does not hold for graph domains which approximately follow the Barabási-Albert model (this includes social network and citation graphs, both common domains for node classification).
> 3. We propose a novel variant of GPN, _LOP-GPN_, which does not make the _irreducibilty of conflicts_ assumption (Section 4). We show how this more computationally expensive model can be trained efficiently (Sections 4.1 and 4.2). Our experimental evaluation shows that the proposed LOP-GPN model significantly outperforms GPN, both in classification accuracy and in quality of the uncertainty estimates. This illustrates the practical importance of the problem we describe.
>
> To answer your question regarding the relation between our work and the baselines used for the experimental analysis, here is a brief explanation for each:
>
> - **APPNP:** This approach is described briefly in Section 3.2 and is used, both, in GPN and LOP-GPN to diffuse information through the graph. By itself, APPNP directly predicts first-order distributions over classes. Therefore, aleatoric (AU) and epistemic uncertainty (EU) cannot meaningfully be distinguished for its predictions. APPNP is included to show the impact of going from first-order to second-order (i.e., GPN and LOP-GPN) predictions while fixing the graph diffusion mechanism. As shown in Table 1, APPNP generally performs either similarly or significantly worse than the "uncertainty-aware" models.
> - **Matérn-GGP:** This model uses a Gaussian Process with a graph Matérn kernel. It models (epistemic) uncertainty via the variance of the posterior distribution at a given vertex. Together with the parameterless GKDE model, Matérn-GGP was included to serve as as a baseline.
> - **GKDE:** The _Graph Kernel Dirichlet Estimate_ is a parameterless model which uses a Gaussian transform of the shortest distance between node pairs $v^{(i)}, v^{(j)}$ as a node kernel $k: \mathcal{V} \times \mathcal{V} \to \mathbb{R}$. A Dirichlet pseudo-count $\alpha^{(i)}_c$ for each class $c$ is defined as one plus the sum over all $k(v^{(i)}, v^{(j)})$, where $v^{(j)}$ are labeled nodes with class $c$. Like GPN, GKDE uses single Dirichlet distributions to model predictive uncertainty. However, unlike GPN, GKDE does not average but _sums up_ pseudo-count estimates of neighbors. The final pseudo-count estimates of GKDE therefore grow with neighborhood size, i.e., epistemic uncertainty goes down as the neighborhood grows. Thus, GKDE only considers a conflict to be irreducible (i.e., high AU **and** EU close to zero) if the number of sampled labeled nodes is sufficiently large. This partially addresses the problem with the _irreducibility of conflicts_ assumption of GPN which we described. Empirically the more flexible LOP-GPN model still clearly outperforms GKDE (see Table 1, Gaussian noise OOD detection results).
>
> We agree that a more detailed explanation of the used baseline models and their relation to our approach is important and we could change our manuscript accordingly.
>
> We hope that this answers your questions and thank you once again for your feedback!

---

### Official Review · Reviewer_GkiD · 2024-03-25

**Q2-1 Originality-Novelty:** 3
**Q2-2 Correctness-Technical Quality:** 3
**Q2-5 Clarity Of Writing:** 2

**Q1 Summary And Contributions:**

As far as I can tell, the paper has three different goals.  The first is philosophical:  the authors aim to provide more precise definitions/analyses of "epistemic uncertainty" and "aleatory uncertainty" respectively; here, the authors use information theory to distinguish the two types of uncertainty.   The second involves a mixture of philosophical and empirical concerns: the authors criticize a particular axiomatic approach to uncertainty quantification, arguing that it's inappropriate in many contexts.

The final task is what I would typically expect to be the focus of a paper in UAI or related conferences.  The authors use propose a new classifier that (mathematically speaking) involves linear pooling of different probability distributions and they evaluate its performance against alternatives.  I say "mathematically speaking" because "linear pooling" is typically used to describe the act of combining multiple experts' beliefs, but in the applications the author has in mind, the probability distributions being combined need not represent anyone's beliefs.  So one could equally say the algorithm takes convex combinations of probability distributions.

In any case, experimental results are provided to show the superiority of the authors' classifier against alternatives.

**Q2-3 Extent To Which Claims Are Supported By Evidence:**

3: Good: the main claims are supported by convincing evidence (in the form of adequate experimental evaluation, proofs, (pseudo-)code, references, assumptions).

**Q2-4 Reproducibility:**

2: Fair: key resources (e.g. proofs, code, data) are unavailable but key details (e.g. proof sketches, experimental setup) are sufficiently well-described for an expert to confidently reproduce the main results.

**Q3 Main Strengths:**

The accuracy of their classifier in the experiments seems impressive.

**Q4 Main Weakness:**

With due respect to the authors, I think the paper is trying to do too much and that two-thirds of the paper -- which deals with what I'd term "philosophical problems" -- could be cut without much loss of content.  In my opinion, only the experimental results in the paper seem sufficiently developed, but given the brevity of the third section of the paper, a reader cannot determine what is being classified in the authors' experiments without consulting Stadler et. al's 2021 paper.

Let me explain.

The paper begins by distinguishing two types of uncertainty (aleatory and epistemic), which are commonly distinguished by philosophers, statisticians, and computer scientists.   The authors then propose to quantify aleatory uncertainty using conditional entropy of a random variable and then to define epistemic uncertainty to be the difference between total entropy and aleatory uncertainty.  That is, the authors propose a philosophical thesis that certain pre-theoretic/qualitative concepts be quantified in a particular way.   But the authors don't provide any philosophical *arguments* for that conclusion.

Such  a philosophical thesis, however, can be defended with arguments.  Consider, for instance, Turing's argument that the pre-theoretic/informal notion of "effectively/mechanical computable" can be identified with "functions computable by a Turing machine" (see Sieg's paper "Church without Dogma", for instance).  Or consider Savage/Anscombe-Aumann/almost every economists' claim that the pre-theoretic notion of "rational action" can be identified with "expected utility maximizer."  Again, such claims can be defended by various arguments (e.g., Savage's theorem).     But as far as I can tell, the authors don't really argue for their philosophical analyses of types of uncertainty.

The argument in the second part of the paper is a bit easier to understand, but it's not clear that the section (and its argument) are necessary at all.  There, the authors criticize a particular "axiomatic" approach to quantifying uncertainty as failing to be inappropriate in certain contexts (I place "axiomatic" in quotation marks because the "axioms" are not mathematically precise as stated).  But one doesn't need to criticize existing approaches to quantifying uncertainty if the goal is simply to motivate an algorithm which one can show outperforms existing algorithms on some metric.

So ultimately, I think the first two sections of the paper that are devoted to these matters are really just a distraction from the novel classifier described in final sections and the experimental results.  I recognize that the authors need to say something about uncertainty quantification before introducing their algorithm, but the first two sections of the paper are too long.  And they take valuable space away from describing and analyzing the experiments in greater detail.

**Q5 Detailed Comments To The Authors:**

See weaknesses.

**Q9 Complying With Reviewing Instructions:**

Yes

---

> ### Author Rebuttal · Authors · 2024-04-02
>
> Thanks for your interesting comments and suggestions.
>
> By way of background, Section 2 gives an overview of the most common approaches to uncertainty quantification in classification tasks, which is relevant for our problem of node classification. Thus, we do not *propose* a quantification of a pre-theoretic/qualitative concept, but only recall what has been proposed in the literature. Besides, in addition to the entropy-based approach in Section 2.1, we also recall an alternative approach in Section 2.2.
>
> Therefore, we also didn't see any need to justify or defend these approaches. Motivations can be found in the references we give. That said, we completely share the reviewer's appreciation of an axiomatic grounding, and we also provide some references where such axiomatic approaches have been studied (beginning of Section 2, point 1). The entropy-based approach origines from information theory, where measures like entropy itself as well as variants and derivates thereof do have an axiomatic justification.
>
> As for the second part, Section 3 provides important background information on GPNs, which is the approach we build on. Without this background, it will be difficult to understand our own method and why it improves the state of the art. We fully agree that the "axioms" should better be called properties or desiderata. We just adopted the terminology from the original paper, but are happy to change this in the final version. Anyway, these properties are important for our motivation: In Section 3.3, we discuss the assumptions in a critical way and explain why other properties might be required.
>
> That said, we do agree that Sections 2 and 3 can be shortened a bit. We could also move Section 3.3 to Section 4, because it is an original contribution and in a sense provides the basis for our own method.
>
> We hope that you find these additional explanations helpful. Once again, thanks a lot for the useful presentation of your point of view.

---

### Meta-Review · Area_Chair_dCrd · 2024-04-16

The authors propose a new approach (LOP-GPN, linear opinion pooled graph posterior network) to quantify uncertainty in semi-supervised node classification. Their approach combines aleatoric and epistemic uncertainty. The reviewers are mostly positive about the paper, even if some are lukewarm.

They appreciate
- an interesting paper on a topic that is very relevant for UAI;
- a well written paper, having a good tutorial value;
- a technically sound approach;
- compelling experimental results showing a high accuracy of LOP-GPN over other approaches.

The weaknesses that they identify are
- a too lengthy digression on the notions of uncertainty, which removes space from the novel contribution on LOP-GPN and the experimental results;
- maybe as a result of the limited space devoted to LOP-GPNs, the novelty is found to be on the incremental side.
- The code was not included at the time of the submission. The authors mention that they intend to publish it algonside with the paper.